# Habituation to a Deterrent Plant Alkaloid Develops Faster in the Specialist Herbivore *Helicoverpa assulta* Than in Its Generalist Congener *Helicoverpa armigera* and Coincides with Taste Neuron Desensitisation

**DOI:** 10.3390/insects13010021

**Published:** 2021-12-23

**Authors:** Dong-Sheng Zhou, Chen-Zhu Wang, Joop J. A. van Loon

**Affiliations:** 1Hunan Key Laboratory for Conservation and Utilization of Biological Resources in the Nanyue Mountainous Region, College of Life Sciences and Environment, Hengyang Normal University, Hengyang 421008, China; zhouds@hynu.edu.cn; 2State Key Laboratory of Integrated Management of Pest Insects and Rodents, Institute of Zoology, Chinese Academy of Sciences, Beijing 100101, China; 3Laboratory of Entomology, Wageningen University, 6700 AA Wageningen, The Netherlands

**Keywords:** habituation, taste desensitization, strychnine, *Helicoverpa armigera*, *Helicoverpa assulta*

## Abstract

**Simple Summary:**

Habituation to deterrent plant compounds has been found in generalist and specialist insect herbivores. The rate at which plant-feeding insects habituate and at which sensitivity of taste neurons detecting deterrents changes has not been compared among closely related species. The generalist *Helicoverpa armigera* (Hübner) and the specialist *Helicoverpa assulta* (Guenée) (Lepidoptera: Noctuidae) differ strongly in degree of host-plant specialism. Both species habituated to the alkaloid strychnine after dietary exposure; however, the specialist *H. assulta* displayed habituation to strychnine faster (at 48 h) than the generalist *H. armigera* (at 72 h). Electrophysiological recordings from taste sensilla on mouthparts revealed that a deterrent-sensitive neuron in the medial sensillum styloconicum of both species displayed significantly reduced sensitivity to the deterrent that coincided in time with the onset of habituation. Our findings show that physiological changes in taste neuron sensitivity coincide with habituation to plant compounds that are initially avoided.

**Abstract:**

The two closely related moth species, *Helicoverpa armigera* and *H. assulta* differ strongly in their degree of host-plant specialism. In dual-choice leaf disk assays, caterpillars of the two species that had been reared on standard artificial diet were strongly deterred by the plant-derived alkaloid strychnine. However, caterpillars of both species reared on artificial diet containing strychnine from neonate to the 5th instar were insensitive to this compound. Fifth instar caterpillars of *H. assulta* and 4th or 5th instars of *H. armigera* not exposed to strychnine before were subjected to strychnine-containing diet for 24 h, 36 h, 48 h, or 72 h. Whereas *H. assulta* displayed habituation to strychnine after 48 h, it took until 72 h for *H. armigera* to become habituated. Electrophysiological tests revealed that a deterrent-sensitive neuron in the medial sensillum styloconicum of both species displayed significantly reduced sensitivity to strychnine that correlated with the onset of habituation. We conclude that the specialist *H. assulta* habituated faster to strychnine than the generalist *H. armigera* and hypothesis that desensitization of deterrent-sensitive neurons contributed to habituation.

## 1. Introduction

The taste system of insects plays an important role in detecting and assessing food-related chemicals. Plant-feeding insect species display remarkable differences in host-plant specialization. Some species feed on plants from many families and are named ‘generalists’, whereas the majority of species feed on plants belonging to a single plant family, genus and often only a single species and are called ‘specialists’. The discrimination of food by insects depends primarily on neural input from their chemosensory systems. In lepidopteran larvae, host-plant discrimination is governed by information from taste neurons that are located primarily on the ventral side of the labrum, the maxillary palps and galea [1,2]. Taste neurons innervating the lateral and medial sensilla styloconica on the maxillary galea play a decisive role in food selection behavior. Former studies have classified the taste neurons in the two sensilla styloconica of several caterpillar species as sugar cell, water cell, salt cell, amino acid cell, inositol cell, deterrent cell, etc. [1]. All herbivorous insects are equipped with deterrent cells, the activation of which leads to an aversive behavioral response [1,3,4,5,6].

Many studies showed that food selection of herbivorous insects is not fixed and can be modified by dietary experiences [7,8,9,10,11,12,13]. It has been documented that a change in food selection behavior normally is associated with changes in taste neuron sensitivity [14,15,16,17]. Taste neuron sensitivity can change adaptively as a result of food experience [16]. Such changes can take two forms: (1) taste neuron sensitivity can be enhanced because the contacted compounds that serve as token stimuli, defined as recognition cues that have a specific botanical occurrence [9,14,15,18]; (2) in contrast, taste neurons may become less sensitive when they have been exposed to deterrent-containing food for a certain period of time [19,20,21,22,23,24]. For example, insects fed a standard artificial diet containing a low level of a feeding deterrent show desensitization of deterrent-sensitive taste neurons [1]. Such reduced sensory responsiveness has been reported in several studies [10,11,12,19,20,21,22,23,24]. Reduced taste neuron sensitivity as a result of dietary history has been reported for both generalist and specialist species [1] and for two phylogenetically closely related specialist species [25]. Here we compared two congeneric *Helicoverpa* species, a specialist and a generalist, focusing on the rate at which habituation and desensitisation develop.

*Helicoverpa armigera* (Hübner) and *Helicoverpa assulta* (Guenée) (Lepidoptera: Noctuidae) are closely related species which exhibit a clearly different degree of host-plant specialism. The former is a generalist feeding on plants from more than 40 families and the latter is a specialist using several species in the family Solanaceae as host plants, such as tobacco and hot pepper [26,27]. In this study, we report the behavior and electrophysiological responses of the two species after experiencing either a standard diet or a diet containing strychnine during varying exposure times. We addressed three questions: (1) Do the two species habituate to the same deterrent? (2) If so, is the rate of habituation similar? (3) Are the onset of habituation and desensitization coinciding? 

## 2. Materials and Methods

### 2.1. Insects

*Helicoverpa armigera* and *H. assulta* were collected from fields in Henan Province, China. We collected the 4th–5th larvae (ca. 200 each time) in July–September every year and these were maintained in the laboratory for the rest of the year. Laboratory colonies of *H. armigera* and *H. assulta* were maintained as continuous cultures in the laboratory in an environmental chamber (about 10 m^2^) under a L16:D8 photo: scotoperiod, at a temperature of 25 ± 1 °C and RH 60–70%. Larvae were reared individually in glass tubes plugged with cotton wool on the standard artificial diet (see below). Adults (30 adults with 1:1 ratio of female to male) were kept in 40 cm × 40 cm × 40 cm wooden cages and supplied with a 10% solution of honey in water. 

### 2.2. Diets

Two types of diets were used: the standard artificial rearing diet for *H. armigera* [28], prepared using the following ingredients (ca. 1 kg): wheat germ (150 g), yeast powder (30 g), methyl-p-hydroxybenzoate (2 g), sorbic acid (1 g); ascorbic acid (3 g), linoleic acid (1 mL), agar (14 g), tomato paste (198 g), and distilled water (600 mL). The rearing diet for *H. assulta* was the same as that of *H. armigera* diet except that chili powder (8 g) was added as a phagostimulant. The experimental diet was the standard diet to which crystalline strychnine was added to obtain a concentration of 0.2 mM. Preliminary tests showed that this concentration allowed sufficiently high survival (78% for *H. armigera* and 75% for *H. assulta*). The chemicals were added to the diet just after heating the agar, and the mixture was stirred vigorously for about 2–3 min [21]. 

### 2.3. Chemicals

Strychnine-HCl was obtained from Sigma-Aldrich (Shanghai, China). Purity of the strychnine was >97%. For behavioral assays, strychnine was diluted in distilled water. For electrophysiological tests, strychnine was diluted in 2 mM KCl, that served as control solution. 

### 2.4. Experimental Design and Behavioral Bioassays

Larvae subjected to behavioral bioassays and electrophysiological tests were reared on standard artificial diet or diet containing strychnine for 5 dietary exposure periods (Figure 1): (a) From neonate to the molt from 4th to 5th instar; (b) 24 h after the molt to the 5th instar; (c) 36 h after the molt to the 5th instar; (d) 48 h after the molt to the 5th instar; (e) 72 h after the molt to the 4th instar. In order to be able to study the effects of 72 h exposure, we used 4th instars for group (e) because 5th instars would have pupated after 72 h. When the results of the tests showed that the caterpillars acquired habituation at a certain time point, we did not test longer exposure times, as we assumed that habituation persisted. Caterpillars for behavioral assays in group (a) were taken at the end of the 4th instar in the non-feeding pre-molt stage and were deprived of food for about 6–8 h after molting to the 5th instar and then tested [29]. Caterpillars of groups (b), (c), (d), and (e) were deprived of food for about 6–8 h after they reached the time points indicated. 

Caterpillars of group (a) were tested to ascertain whether *H. armigera* and *H. assulta* exhibited habituation after long term exposure to strychnine. Caterpillar groups (b)–(e) served to investigate how fast *H. armigera* and *H. assulta* habituated when exposed to the deterrent in either the 5th (groups b, c, and d) or 4th instar (group e). Pepper fruit disks were used as the substrate in dual-choice disk tests. Pepper (*Capsicum frutescens* L. (Solanaceae), cultivar ‘JingTian-5’) was grown in the greenhouse of Daxing Agricultural Station in Beijing. For each bioassay, 20 replications were performed. Strychnine-HCl was diluted in distilled water immediately prior to application. Five microliters of solution of the test compounds was applied on a disk cut from pepper fruit (thickness 1 mm, surface area 78.5 mm^2^). Taking the disk fresh weight into account, the final concentration of strychnine tested was 2 mM, assuming homogeneous distribution through the disk volume. Advantages of pepper fruit disks are that the solutions are absorbed evenly by the fruit tissue and do not leach out. The experiments were carried out in a climatic chamber at a temperature of 25 ± 1 °C. Larvae were placed individually in the middle of Petri dishes lined with moist filter paper. Four treated and four control disks were alternately arranged in an ABABABAB fashion around the circumference of the Petri dishes. This arrangement was modified relative to earlier studies to ensure that larvae had an equal chance of encountering both treated and control disks [29,30]. Consumption was observed at 1-h intervals thereafter. When ca. 50% of the control disks or treated disks had been eaten, the disk remains were digitally scanned using a Hewlett-Packard flatbed scanner. Disk surface area was measured using Scion Image (Scion Co., Frederick, MD, USA) (freeware, https://ittechgyan.com/download-scion-image-software/, accessed on 14 December 2021). The areas consumed were calculated by subtracting the remaining areas of leaf disks from the average area of 3 reference disks which served as shrinkage controls [31].

### 2.5. Electrophysiology

For each species and every duration of dietary exposure ((a)–(e)), there were two groups of larvae to be tested: caterpillars reared on standard artificial diet and those reared on the diet containing strychnine. The concentrations of strychnine tested were 0, 0.01, 0.1, 1, 10 mM. The tip recording technique [32] was used to record responses to the different stimuli from the sensilla styloconica on the maxillary galea. Fifth instar caterpillars were used for electrophysiological tests and were starved for 2 h before testing [33]. Group (e) *H. armigera* caterpillars were subjected to recording just after they had molted to the 5th instar. Excised caterpillar heads were mounted on a silver wire electrode which was connected to the input of a pre-amplifier (Syntech Taste Probe DTP-1, Hilversum, The Netherlands). Stimulus solutions were filled into glass micropipettes with a tip diameter of c. 30 µm. Amplified signals were digitized by an A/D-interface (Syntech IDAC-4, Hilversum, The Netherlands) and sampled into an Intel Pentium-based personal computer. Electrophysiological responses were quantified by counting the number of spikes in the first 1010 ms after the start of stimulation that typically lasted 3–4 s; the first 10 ms contained the contact artifact and was skipped for analysis. Spikes were counted visually by the experimenter with the aid of Autospike version 3.7 software (Syntech, Hilversum, The Netherlands).

### 2.6. Statistical Analysis

A feeding deterrent index (FDI) = 100(C − T)/(C + T) was calculated to quantify strength of deterrence, where C represents the fruit disk area consumed from control disks, T represents the fruit disk area consumed from treated disks. Paired *t*-tests were performed to analyze the disk consumption data. In electrophysiological tests, standard diet-reared caterpillars and deterrent-exposed caterpillars are independent samples; thus, a two sample *t*-test was performed to compare electrophysiological responsiveness of insects reared on different diets. All statistical analyses were conducted using SPSS 13.0 (SPSS Inc., Chicago, IL, USA). *p* values < 0.05 were considered significant.

## 3. Results

### 3.1. Behavioral Bioassays

*Helicoverpa armigera* and *H. assulta* larvae exposed to the standard diet from neonate until the molt to the 5th instar (group a) were strongly deterred by strychnine (FDI = 40.5 and 50, respectively; *p* < 0.01; Figure 2A). However, *H. armigera* and *H. assulta* caterpillars exposed to diet containing strychnine from neonate until the molt to the 5th instar did not discriminate between pepper disks with strychnine and control disks (FDI = 4.7 and 7.1, respectively; *p* > 0.05; Figure 2A).

*Helicoverpa armigera* 5th instar caterpillars exposed to standard artificial diet that were raised in parallel to the strychnine-exposed groups (b)–(e), were sensitive to strychnine (FDI = 46.3, 41.5, 45.2, and 42.3, respectively; *p* < 0.01; Figure 2B–E). Similarly, *H. armigera* caterpillars which had been feeding for 24 h, 36 h, and 48 h during the 5th instar on diet containing strychnine were also sensitive to strychnine (FDI = 43.4, 39.4, and 30.5, respectively; *p* < 0.01, *p* < 0.01, and *p* < 0.05, respectively; Figure 2B–D). However, *H. armigera* 4th instar caterpillars that had experienced diet containing strychnine for 72 h did not distinguish the pepper disks treated with strychnine from the control (FDI = 10.6; *p* > 0.05; Figure 2E).

Caterpillars of *H. assulta* that had experienced standard artificial diet for 24 h, 36 h and 48 h since the molt to the 5th instar (groups (b)–(d)) were highly sensitive to strychnine (FDI = 51, 49.6, and 49.2, respectively; *p* < 0.01; Figure 2B–D). Likewise, *H. assulta* caterpillars that had experienced diet containing strychnine for 24 h and 36 h were also sensitive to strychnine (FDI = 32.8 and 26.4, respectively; *p* < 0.05; Figure 2B,C). However, *H. assulta* caterpillars that had been feeding on diet containing strychnine for 48 h did not discriminate the disks treated with strychnine from related control disks (FDI = 12.5, *p* > 0.05; Figure 2D).

### 3.2. Electrophysiological Responses

No electrophysiological response was recorded from the lateral sensilla styloconica of *H. assulta* when stimulated with strychnine in the dose-range tested; strychnine elicited a weak response from the lateral sensilla styloconica of *H. armigera* [11]. Strychnine elicited a strong response from the medial sensilla styloconica of both *H. armigera* and *H. assulta* caterpillars (Figure 3A and Figure 4B,D). The deterrent neuron in the medial sensilla styloconica of both *H. armigera* and *H. assulta* caterpillars which had been feeding on standard artificial diet was significantly more sensitive to strychnine than the equivalent neuron of caterpillars exposed to diet containing strychnine at 0.1, 1 and 10 mM (*H. armigera*: *p* < 0.05, Figure 3A; *H. assulta*: *p* < 0.05, *p* < 0.01, and *p* < 0.01, respectively, Figure 3A). No significant difference was found between the sensitivity of the deterrent neuron in the medial sensilla styloconica of *H. armigera* caterpillars fed on standard artificial diet and those fed on strychnine-containing diet for 24 h, 36 h, and 48 h into the 5th instar (*p* > 0.05, Figure 3B–D). However, when *H. armigera* 5th instar caterpillars had been exposed to strychnine-containing diet for 72 h, i.e., the entire duration of the 4th instar, they produced a significantly weaker response to 1 and 10 mM strychnine in the medial deterrent neuron than caterpillars fed on standard artificial diet during the same period (*p* < 0.05; Figure 3E). Similarly, only *H. assulta* caterpillars that had experienced strychnine-containing diet for 48 h produced a significantly weaker response to 1 and 10 mM strychnine in medial sensilla styloconica than caterpillars that had been feeding on standard artificial diet during the same period (*p* < 0.05; Figure 3D).

### 3.3. Relationship between Electrophysiological and Behavioral Responses

To examine the relationship between electrophysiological and behavioral responses, these have been plotted against each other (Figure 5). As the duration of exposure increases, both the electrophysiological and behavioral responsiveness diminish, seen as shifts to lower values along the horizontal and vertical axes of the plot in both species. The difference between the two species is that the shift to habituation (values of FDI not significantly different from zero) occurs between 36 and 48 h of exposure in *H. assulta* (Figure 5B), whereas it occurs between 48 and 72 h in *H. armigera* (Figure 5A). 

## 4. Discussion

Dual-or multiple-choice tests have commonly been performed to assess behavioral discrimination between plants [16]. In this study, we used a dual-choice test to evaluate the deterrent effect of strychnine. We selected pepper fruit tissue as the test substrate in behavioral assays because pepper is an acceptable plant for both species, and the solution of strychnine was readily taken up by the fruit disks.

Habituation to deterrent chemicals has been demonstrated in many cases in the laboratory. For example, recent research reported larvae of the generalist herbivore *Agrotis ipsilon* (Hufnagel) (Lepidoptera: Noctuidae) acquired habituation when they were exposed to the three chemically diverse deterrents strychnine, chlorogenic acid, and fumaropimaric acid [12]. Previous studies also showed that *P. rapae* caterpillars fed on cabbage (*Brassica oleracea* L.; one of its favorite host plants) refused to feed on nasturtium (*Tropaeolum majus* L.; an acceptable but less preferred host plant), and this rejection behavior was ascribed to the presence of the deterrent phenolic chlorogenic acid. However, *P. rapae* caterpillars reared on nasturtium from neonate onward were desensitized to chlorogenic acid [8,34]. When *P. rapae* was reared on artificial diet for its entire larval development, its behavioral discrimination of the chemically diverse deterrents chlorogenic acid, naringin and strychnine was significantly reduced [10]. Dietary experience influenced food selection behavior of other plant-feeding insect species [3,7,9,35,36,37,38]. Our previous experiments showed that *H. armigera* caterpillars habituated to strychnine, either when presented on cotton leaf disks or on pepper fruit disks, after feeding from neonate to the 5th instar on an artificial diet containing strychnine [11]. Strychnine is an alkaloid found in the seeds of the *Strychnos nux-vomica* L. tree, which was proved in previous studies to be a deterrent for several insects among which gypsy moth larvae (*Lymantria dispar* L.) that has a taste neuron in its medial maxillary styloconic sensillum that responds robustly to this alkaloid [39]. Strychnine activates a bitter-sensitive neuron in I-type sensilla in *Drosophila melanogaster* (Meigen) (Diptera: Drosophilidae) [40]. Strychnine is not found in the recorded host plants of *H. armigera* and *H. assulta* and turned out to represent a strong deterrent for caterpillars reared on the standard artificial diet in the present study (Figure 2A).

Group (a) caterpillars of both species habituated to strychnine after exposure from neonate to the molt to the 5th instar (Figure 2A). Group (b)–(d) *H. armigera* caterpillars that had experienced strychnine for 24 h, 36 h, 48 h since the molt to the 5th instar preferred to feed from control disks over disks treated with strychnine; however, after *H. armigera* 4th instar caterpillars fed on diet containing strychnine for 72 h, habituation to strychnine was found (Figure 2E). This suggests that habituation occurred between 48 h and 72 h of exposure. For *H. assulta*, our data showed that this specialist feeder habituated to strychnine after they experienced strychnine-containing diet for 48 h into the 5th instar (Figure 2D). Although we cannot exclude that the rate at which habituation develops differs between the 4th and 5th instar, we conclude that the specialist *H. assulta* habituated to strychnine faster than the generalist *H. armigera*. This finding is unexpected as we hypothesized that the generalist *H. armigera* would habituate faster to secondary plant compounds. This hypothesis was based on studies on two *Heliothis* species, taxonomically closely related to *Helicoverpa*, demonstrating that the specialist species had greater sensitivity to deterrents than the generalist [4,41]. However, these studies focused on short-term (3 min) behavioral responses of the two *Heliothis* species. A major difference with our study is the chronic exposure to the deterrent that resulted in habituation. This suggests that the sensitivity level of naïve individuals, which we also found to be higher in the specialist *H. assulta* (Figure 4B,D), does not predict rate of habituation.

It has been documented that insects fed a standard artificial diet containing a low level of a feeding deterrent show desensitization of receptors responding to these compounds [1]. For example, *Manduca sexta* fed an artificial diet containing salicin, caffeine or aristolochic acid had reduced taste neuron sensitivity to the corresponding chemicals [21]. Maxillary taste neurons of *P. brassicae* caterpillars were less sensitive to chlorogenic acid when reared on an artificial diet than when reared on cabbage [24]. The closely related species *P. rapae* acquired desensitization in an analogous deterrent neuron in medial sensilla styloconica to chlorogenic acid, naringin, and strychnine upon long-term exposure to a similar artificial diet [10]. Our electrophysiological data on *H. armigera* and *H. assulta* are in line with the findings on *Manduca* and *Pieris*. Electrophysiological tests on caterpillars exposed to strychnine from neonate to 5th instar showed that the deterrent neuron in medial sensilla styloconica of both species expressed a significantly reduced sensitivity to strychnine. Interestingly, by testing varying exposure times, the electrophysiological desensitization of the deterrent neuron in medial sensilla styloconica of both species showed a remarkable coincidence in time with the estimated onset of habituation: for *H. armigera*, the deterrent neuron became significantly less sensitive to strychnine after the caterpillars experienced strychnine diet for 72 h; and the deterrent neuron in medial sensilla styloconica of *H. assulta* also had significantly reduced sensitivity to strychnine after they fed on strychnine diet for 48 h. The sensitivity of taste neurons is not fixed and can change in response to dietary history, suggesting that taste cells possess a “peripheral memory” [1]. In *Drosophila*, sensory neurons play a very important role in state-dependent gain control of behavior [42]. A recent study showed that *H. armigera* and *H. assulta* adapted to toxic secondary metabolites at gustatory levels, which suggests the taste neurons play an important role in adaptation [43]. Our electrophysiological tests showed that the diet-induced habituation to deterrents can at least partly be explained by chemosensory desensitization of the deterrent neuron. Our results do not exclude that changes have occurred simultaneously in the central nervous system. It has been suggested that the central nervous system plays a part in habituation [18,44]. However, the coincidence between chemosensory desensitization and habituation in our case suggests that peripheral changes play a prominent role in the causation of behavioral habituation. This is in agreement with findings on the black cutworm, *A. ipsilon**,* showing that the same deterrent taste neuron desensitized through dietary exposure was correlated with cross-habituation to two structurally different chemicals [12]. The taste system could contribute to behavioral plasticity by amplifying or attenuating taste input at the level of the central nervous system or directly at the level of the gustatory receptor neurons, which could modulate the level of expression of their different receptor proteins [45,46]. We hypothesize taste neuron desensitization may be ascribed to down-regulation of expression of gustatory receptor (GR)-proteins tuned to strychnine and/or to down-regulation of intracellular signal-transduction cascades. Additional experiments employing gene expression analysis are required to test these hypotheses.

Although our results showed the specialist *H. assulta* habituated to the alkaloid deterrent faster than the generalist *H. armigera* and the desensitization of the deterrent neuron in medial sensilla styloconica offers a physiological explanation for the habituation process, there are a number of limitations to our study that should be noted. First, we only used one deterrent for this comparative study. However, studies showed that when insects habituated to one deterrent, “generalization” [47,48] or “cross-habituation” [8,10,12] can occur, even to molecules of chemically quite diverse classes, such as phenolics and steroids. Second, we confined the electrophysiological tests to the maxillary sensilla styloconica, and, although these play a decisive role in food selection behavior, caterpillars possess additional gustatory sensilla which could mediate detection of deterrents, such as epipharyngeal sensilla [1] and sensilla on the maxillary palp tip [49]. Third, in this study, we focused on the peripheral neural system. However, habituation resulting from dietary history is most likely based on the combined influence of physiological changes in central nervous pathways and peripheral chemosensory desensitization.

The behavioral and electrophysiological results taken together, we conclude that (1) the specialist *H. assulta* habituated to the deterrent strychnine faster than the generalist *H. armigera*; (2) desensitization of the deterrent neuron in medial sensilla styloconica of both species contributes to the habituation observed; and (3) the coincidence between the behavioral habituation and chemosensory desensitization in both species suggests that peripheral changes play a significant role in the causation of habituation.

## Figures and Tables

**Figure 1 insects-13-00021-f001:**
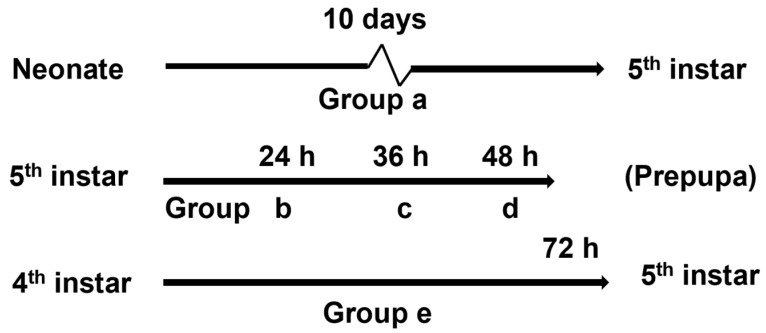
The five exposure protocols to assess habituation of *H. armigera* and *H. assulta* upon exposure to the deterrent alkaloid strychnine tested in this study.

**Figure 2 insects-13-00021-f002:**
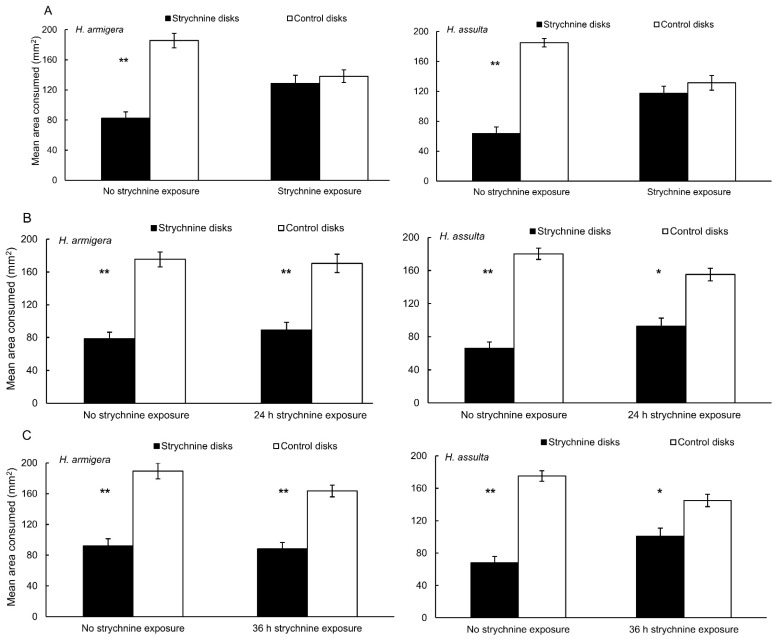
Feeding preference behavior of *H. armigera* larvae and *H. assulta* larvae, reared on standard artificial diet (no strychnine exposure) or artificial diet containing strychnine for different durations of dietary exposure, indicated along horizontal axis, on pepper fruit disks in choice assays between control disks and disks containing strychnine. Left lane: *H. armigera* larvae; right lane: *H. assulta* larvae. (**A**) Exposed from neonate until the molt to the 5th instar; (**B**) exposed during 24 h since the molt to the 5th instar; (**C**) exposed during 36 h since the molt to the 5th instar; (**D**) exposed during 48 h since the molt to the 5th instar; (**E**) *H. armigera* larvae, exposed during 72 h since the molt to the 4th instar; replicated 20 times for each assay. Vertical lines represent standard errors. Asterisks indicate significant differences between treated and control disks according to the paired-samples *t*-test (* *p* < 0.05, ** *p* < 0.01).

**Figure 3 insects-13-00021-f003:**
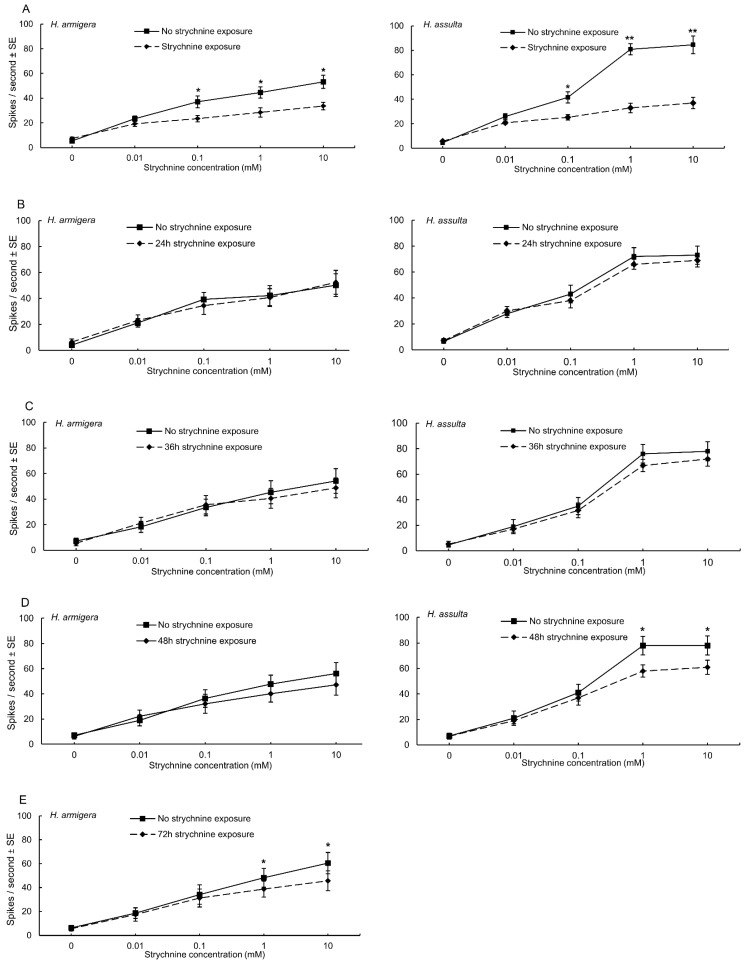
Electrophysiological dose-response curves for strychnine; spike frequencies of the deterrent neuron in the medial styloconic sensilla of *H. armigera* and *H. assulta* caterpillars reared either on standard artificial diet (squares, no strychnine exposure) or artificial diet containing strychnine (diamonds) for the intervals indicated. (**A**) Exposed from neonate until the molt to the 5th instar *H. armigera* larvae (*n* = 10) and *H. assulta* larvae (*n* = 11); (**B**) exposed during 24 h since the molt to the 5th instar; *H. armigera* larvae (*n* = 11) and *H. assulta* larvae (*n* = 9); (**C**) exposed during 36 h since the molt to the 5th instar; *H. armigera* larvae (*n* = 10) and *H. assulta* larvae (*n* = 10); (**D**) exposed during 48 h since the molt to the 5th instar; *H. armigera* larvae (*n* = 10) and *H. assulta* larvae (*n* = 10). (**E**) *H. armigera* larvae, exposed during 72 h since the molt to the 4th instar; *n* = 10. Vertical lines represent standard errors. Asterisks indicate a significant difference in response frequency between different groups of diet-experienced caterpillars (two-sample *t*-test, * *p* < 0.05; ** *p* < 0.01).

**Figure 4 insects-13-00021-f004:**
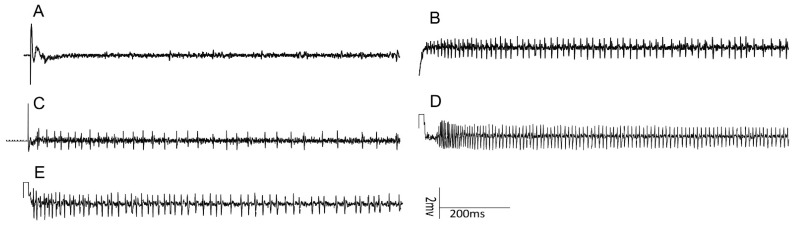
Exemplary recordings of electrophysiological activity in medial sensilla styloconica on the maxillary galea of 5th instar *H. armigera* and *H. assulta* caterpillars reared on standard artificial diet or on artificial diet containing strychnine: (**A**) control (KCl 2 mM); (**B**) strychnine 1 mM on *H. armigera* caterpillars reared on standard artificial diet; (**C**) strychnine 1 mM on *H. armigera* caterpillars exposed to strychnine-containing diet for 72 h; (**D**) strychnine 1 mM on *H. assulta* caterpillars reared on standard artificial; (**E**) strychnine 1 mM on *H. assulta* caterpillars reared on artificial diet containing strychnine for 48 h. The onset of the stimulations occurred at the start of each trace.

**Figure 5 insects-13-00021-f005:**
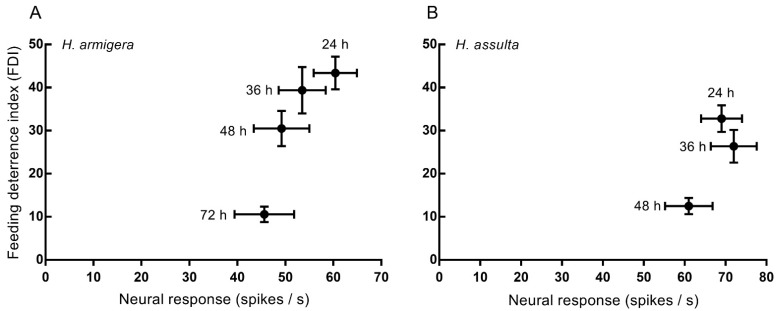
Relationship between electrophysiological and behavioral responses in *H. armigera* and *H. assulta caterpillars*. Plot of feeding deterrent index (FDI) as a function of electrophysiological response strength upon stimulation with a 10 mM solution of strychnine in (**A**) *H. armigera*, (**B**) *H. assulta* caterpillars exposed for the durations indicated (24 h, 36 h, 48 h, or 72 h). Control solution was 2 mM KCl. Errors bars represent s.e.m.

## Data Availability

The data presented in this study are available on request from the first author.

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
