# Peer review of "Habituation to a Deterrent Plant Alkaloid Develops Faster in the Specialist Herbivore Helicoverpa assulta Than in Its Generalist Congener Helicoverpa armigera and Coincides with Taste Neuron Desensitisation"

_insects, 2021, doi:10.3390/insects13010021_

Round 1
Reviewer 1 Report
Zhou et al present clear data indicating that Helicoverpa assulta larvae habituate to strychnine more quickly than H armigera larvae. This faster habituation seems correlated with the smaller diet breath of H assulta as compared to H armigera. This study follows a previous paper of the same authors where they have examined the impact of exposure to strychnine and strophantin-K on H armigera larvae.
The data supporting these conclusions are behavioral observations (feeding choices) and electrophysiological recordings (spiking frequencies responses to strychnine from one taste sensillum, the medial sensillum styloconica (MSS)). The authors show that (1) rearing larvae on a food medium mixed with strychnine decreases the aversiveness of this molecule and also decreases the sensitivity of MSS toward strychnine in L5 larvae and (2) the decrease of sensitivity is observed in L5 larvae depends on the exposure duration.
The data are clean and should provide a sound basis for further exploration of the mechanisms involved in such a decrease of sensitivity toward strychnine. The authors may cite and discuss the work of Akhtar and Isman (2003-2004) who showed comparable observations on Trichoplusia ni. Habituation may involve changes in the sensory detection, in the choices made at the level of the brain, but also may involve changes in terms of the expression of detoxification enzymes.
Minor:
I don’t understand why the authors have tested strychnine, besides the fact that strychnine works. Is there any ecological reason behind testing strychnine? Is it a compound to which these 2 species are exposed in nature? Is strychnine toxic to the larvae?
The figures are ok but complicated to read and to grasp and to summarize. Presenting the behavioral data and electrophysiological data separately does not help to see the correlation. A summarizing figure with 1 axis corresponding to the behavior and 1 axis with the electrophysiology data may help. The authors plot the mean and sem. The trend now is to plot also individual points, especially when dealing with small experimental series.
Line 84 “were maintained”: how long were these species maintained in the laboratory? Herbivorous species maintained in the laboratory may quickly change their behavior and become less responsive.
Line 97: the strychnine concentration chosen is 0.2 mM. This concentration is chosen because “preliminary tests” showed it allowed 70% survival. Is the survival of the larvae the same in the 2 species? I would expect H assulta to be more sensitive to xenobiotics than H armigera?
Line 107: a general diagram of the exposure procedures might help. The reader has no idea about the duration of each larval stage. Nor how the experimenter achieve to identify when the larvae are moulting. More explanations regarding how the larvae are followed would be helpful. My understanding of how these larvae are growing is that there might be considerable variability in the duration of each stage depending on the individuals. It would be also useful to have information on the duration of the development in the 2 species when they are exposed to strychnine as compared to when they are not exposed – the presence of toxins or antifeedants may increase considerably the development time.
Lines 136-138: it is not clear to me when the measures were done. The text says “when 50% of either of the disks” are eaten. This suggests the data are collected at different times depending on the experiment and the concentration tested?
Line 139: the internet link dots to a non-existent web site. It is thus impossible to readers to get this program.
Author Response
Zhou et al present clear data indicating that Helicoverpa assulta larvae habituate to strychnine more quickly than H armigera larvae. This faster habituation seems correlated with the smaller diet breath of H assulta as compared to H armigera. This study follows a previous paper of the same authors where they have examined the impact of exposure to strychnine and strophantin-K on H armigera larvae.
The data supporting these conclusions are behavioral observations (feeding choices) and electrophysiological recordings (spiking frequencies responses to strychnine from one taste sensillum, the medial sensillum styloconica (MSS)). The authors show that (1) rearing larvae on a food medium mixed with strychnine decreases the aversiveness of this molecule and also decreases the sensitivity of MSS toward strychnine in L5 larvae and (2) the decrease of sensitivity is observed in L5 larvae depends on the exposure duration.
The data are clean and should provide a sound basis for further exploration of the mechanisms involved in such a decrease of sensitivity toward strychnine. The authors may cite and discuss the work of Akhtar and Isman (2003-2004) who showed comparable observations on Trichoplusia ni. Habituation may involve changes in the sensory detection, in the choices made at the level of the brain, but also may involve changes in terms of the expression of detoxification enzymes.
Reply: We thank the reviewer for the suggestions made. We included citations to the relevant papers of Akhtar et al. (2003) and Akhtar and Isman (2004) in the revised manuscript, in the Introduction and Discussion respectively.
Minor:
I don’t understand why the authors have tested strychnine, besides the fact that strychnine works. Is there any ecological reason behind testing strychnine? Is it a compound to which these 2 species are exposed in nature? Is strychnine toxic to the larvae?
Reply: The choice for strychnine was motivated by our previous work that demonstrated that it is an effective deterrent, also noted by the reviewer. In this sense it can be considered as a model deterrent compound that has been used as such in many studies on the deterrency of secondary plant compounds to plant-feeding insects (see review by Schoonhoven and Van Loon, 2002 [1]). Regarding possible toxicity to the larvae, preliminary tests showed 0.2 mM strychnine diet allowed sufficiently high survival rates for both species (see below) when long-term exposure started in the neonate stage, indicating mild toxicity.
The figures are ok but complicated to read and to grasp and to summarize. Presenting the behavioral data and electrophysiological data separately does not help to see the correlation. A summarizing figure with 1 axis corresponding to the behavior and 1 axis with the electrophysiology data may help. The authors plot the mean and sem. The trend now is to plot also individual points, especially when dealing with small experimental series.
Reply: We thank the reviewer for the constructive suggestion. We added a new figure (Fig. 5) plotting the behavioural response as function of the electrophysiological response that shows the concurrent shifts in both behavioural and chemosensory sensitivity to the deterrent.
Line 84 “were maintained”: how long were these species maintained in the laboratory? Herbivorous species maintained in the laboratory may quickly change their behavior and become less responsive.
Reply: We collected the 4th-5th larvae in the fields in July-September each year and then maintained these in the laboratory for the rest of the year.
Line 97: the strychnine concentration chosen is 0.2 mM. This concentration is chosen because “preliminary tests” showed it allowed 70% survival. Is the survival of the larvae the same in the 2 species? I would expect H assulta to be more sensitive to xenobiotics than H armigera?
Reply: Survival of the larvae was similar for the two species: 78% for H. armigera and 75% for H. assulta). These numbers have been included in the revision in Materials and methods.
Line 107: a general diagram of the exposure procedures might help. The reader has no idea about the duration of each larval stage. Nor how the experimenter achieve to identify when the larvae are moulting. More explanations regarding how the larvae are followed would be helpful. My understanding of how these larvae are growing is that there might be considerable variability in the duration of each stage depending on the individuals. It would be also useful to have information on the duration of the development in the 2 species when they are exposed to strychnine as compared to when they are not exposed – the presence of toxins or antifeedants may increase considerably the development time.
Reply: We provided the information requested by adding a new figure (Fig. 1) to visualise the habituation protocol in the revised manuscript. We observed that development of the two species exposed to strychnine was only slightly slower than that observed for the non-exposed controls, in line with the mild toxicity mentioned above.
Lines 136-138: it is not clear to me when the measures were done. The text says “when 50% of either of the disks” are eaten. This suggests the data are collected at different times depending on the experiment and the concentration tested?
Reply: The duration of the behavioural tests was 4-6 hours. Each assay was terminated when ca. 50% of the control disks or treated disks had been eaten, then the remaining disk areas were digitally scanned using a Hewlett-Packard flatbed scanner. The T50 criterion is generally applied in the literature, the argument being that the choice may be affected when the total surface area of one the two leaf disk types (e.g. the control disks) is becoming much lower than that of the other because the likelihood to encounter it is skewed.
Line 139: the internet link dots to a non-existent web site. It is thus impossible to readers to get this program.
Reply: We thank the reviewer for pointing this out. We inserted an updated link in the revised manuscript.
Reviewer 2 Report
L16-17: I disagree with this sentence: previous papers have shown differences in sensitivity towards bitter compounds in two closely related species (i.e. P hospiton and P machaon)
L 24-25: Recent studies on P hsopiton caterpillars and adults have shown that not only the different activation of the deterrent cells affects the host plant choice the larvae, but it can also lead to a divergence with the choices of the ovipositing females.
L 63-64: it would be advisable for the authors to add some references
L 67-69: Authors should consider all literature: papers have already been published comparing taste sensitivity in closely related species and their peripheral plasticity
L 83: in which period of the year? how many specimens were collected? what stage did they have? Were only field or also farm animals used for the experiments?
L 85: what was the size of the environmental chamber?
L 87-88: how many adults were in the cage?
L 91: why didn't the authors reared the larvae on their host plant?
L 97-98: who were these preliminary tests conducted by? I find 30% mortality to be quite high.
L 111: 3 days from moulting to the fifth stage to pupa seems to me a very short time. Are all stages that short? Again: since the fifth stage did not allow to arrive at 72h, why not do all the experiments at the fourth stage? In fact, sensory plasticity decreases with age.
L 111-113: let me understand: as soon as the caterpillar began to eat the diet supplemented with strychnine, was it removed from the experimental arena?
L 114: I believe that "and" after (a) is to be removed
L 119-121: however, the larvae of group (e) were in the 4th stage: this passage should be better explained
L 127: how do the authors assume this homogeneous distribution?
L 145-146: why strychnine was not tested at the 2mM concentration in analogy with the behavioral tests?
Electrophysiological experiments: how long did a stimulation last? Did the authors have a control solution?
Statistical analysis: which data were considered significant?
L 168-172 - Fig. 1: I find it difficult to follow the text and the representation in the figure: they should be more aligned
L 190: I understand that since H assulta already adapted to 48h, the experiments were not performed at 72h. I find this procedure a bit anomalous. Wouldn't it have been interesting to see if the degree of habituation changed?
Fig. 2D: it would be advisable for the authors to be more precise and uniform in the graphical representation of the data
Fig. 3: I think the authors considered the initial contact artifact and started counting spikes in the second following this artifact. But it is not reported in the M&M
L 246: I think that the electrode/sensillum tip contact and therefore the start of the stimulation precedes the start of the neural trace. Also, I don't understand why the authors do not show the neural traces of both species and at all tested concentrations
L 263: In P. machaon it was shown that the diet changed its sensory response profile and also the neural code used for discrimination, making it more similar to that of P hospiton and less to that of P. machaon raised on a different diet. But this is not discussed by the authors
L 279-280: I don't think the larvae discriminate in this case. Authors should rewrite the sentence
L 290-291: Similar results were found in P. hosption, a practically monophagous species, compared to the polyphagous P. machaon, with which they are closely related.
L 331-334: this concept is very interesting, but it should be better argued, perhaps with some references
L 345: I do not agree with this: in fact in the maxillary palp the greatest number of sensilla are mainly olfactory
Author Response
L16-17: I disagree with this sentence: previous papers have shown differences in sensitivity towards bitter compounds in two closely related species (i.e. P hospiton and P machaon)
Reply: We have rephrased the sentence and have referred to the comparison between the two Papilio species.
L 24-25: Recent studies on P hsopiton caterpillars and adults have shown that not only the different activation of the deterrent cells affects the host plant choice the larvae, but it can also lead to a divergence with the choices of the ovipositing females.
Reply: The relationship between larval and adult chemosensory responses the reviewer refers to is outside of the scope of this paper.
L 63-64: it would be advisable for the authors to add some references
Reply: We have inserted references (19-24).
L 67-69: Authors should consider all literature: papers have already been published comparing taste sensitivity in closely related species and their peripheral plasticity
Reply: We have inserted two references on taste sensitivity in closely related species and their peripheral plasticity (17, 25).
L 83: in which period of the year? how many specimens were collected? what stage did they have? Were only field or also farm animals used for the experiments?
Reply: We collected the 4th-5th larvae (around 200 each time) in the fields in July-September every year and then maintained these in the laboratory for the rest of the year. We used laboratory colonies of both caterpillars in the experiments. We have added these details in the revised manuscript.
L 85: what was the size of the environmental chamber?
Reply: The environmental chamber was around 10 m2. We have added the details in the revised manuscript.
L 87-88: how many adults were in the cage?
Reply: We put 30 adults with 1:1 ratio of female to male and have added these details in the revised manuscript.
L 91: why didn't the authors reared the larvae on their host plant?
Reply: The reason for this was that the host plants of both species are not available the whole year round and both species have been routinely reared on artificial diet in our lab.
L 97-98: who were these preliminary tests conducted by? I find 30% mortality to be quite high.
Reply: The first author performed the preliminary tests. We consider 22 – 25% mortality acceptable.
L 111: 3 days from moulting to the fifth stage to pupa seems to me a very short time. Are all stages that short? Again: since the fifth stage did not allow to arrive at 72h, why not do all the experiments at the fourth stage? In fact, sensory plasticity decreases with age.
Reply: In our experience the fifth stage to pupa of the two Helicoverpa species under the rearing conditions they were kept lasts 3 – 5 days, however, most of the caterpillars stop feeding after 3 days and prepare to pupate. The earlier instars last even shorter: L1 to L2 ca. 2 days, L2 to L3 ca. 2 days, L3 to L4 3 days, L4 to L5 took about 3 days. We carried out the behavioural experiments on both the fourth and the fifth instars as described. Electrophysiological tests were done on fifth instars due to the higher success rate of tip-recording from larger sized sensilla.
L 111-113: let me understand: as soon as the caterpillar began to eat the diet supplemented with strychnine, was it removed from the experimental arena?
Reply: The interpretation of the reviewer is not correct: the behavioural assay lasted 4 – 6 hours, as described in the text (see above).
L 114: I believe that "and" after (a) is to be removed
Reply: Deleted.
L 119-121: however, the larvae of group (e) were in the 4th stage: this passage should be better explained
Reply: We rephrased the sentence to explain that group (e) larvae of H. armigera were recorded from just after they moulted to the 5th instar.
L 127: how do the authors assume this homogeneous distribution?
Reply: One of the advantages of the pepper fruit disks is that the solution is quickly absorbed by the cut fruit tissue and we assumed that it was evenly distributed.
L 145-146: why strychnine was not tested at the 2mM concentration in analogy with the behavioral tests?
Reply: We agree that it would have been more consistent to test the deterrent at 2 mM, since this was the concentration assumed to be present in the pepper fruit disks (see also previous reply). However, an unknown level of variation in concentration of the deterrent in the fruit disk tissue cannot be excluded; this is the reason why we tested a wider range of concentrations in the electrophysiological experiments.
Electrophysiological experiments: how long did a stimulation last? Did the authors have a control solution?
Reply: Stimulations typically lasted for 3 - 4 s. We have added this in Materials and methods. For electrophysiological tests strychnine was diluted in 2 mM KCl, that served as control solution, as stated in Materials and methods.
Statistical analysis: which data were considered significant?
Reply: P < 0.05 was considered significant. We have added this in revised manuscript.
L 168-172 - Fig. 1: I find it difficult to follow the text and the representation in the figure: they should be more aligned
Reply: We added that the first results described in these lines referred to group (a) which we think clarifies the text.
L 190: I understand that since H assulta already adapted to 48h, the experiments were not performed at 72h. I find this procedure a bit anomalous. Wouldn't it have been interesting to see if the degree of habituation changed?
Reply: We agree that it would have been interesting to see what happened after 48 h for H. assulta. As our aim is to compare the rate of the habituation between the two species we stopped when habituation occurred.
Fig. 2D: it would be advisable for the authors to be more precise and uniform in the graphical representation of the data
Reply: We have modified the panel of Fig. 2D to have the same format as the other panels.
Fig. 3: I think the authors considered the initial contact artifact and started counting spikes in the second following this artifact. But it is not reported in the M&M
Reply: The first 10 ms was skipped since it contained the stimulus onset artifact. We have added this in Materials and methods.
L 246: I think that the electrode/sensillum tip contact and therefore the start of the stimulation precedes the start of the neural trace. Also, I don't understand why the authors do not show the neural traces of both species and at all tested concentrations
Reply: We showed exemplary recordings of electrophysiological activity in medial sensilla styloconica of both species reared on standard diets or diets containing strychnine. We selected 1 mM concentration to show the neural responses of non-exposed and habituated caterpillars and think that these allow a reliable impression of differences. Adding more traces would occupy a disproportional amount of space in the paper compared to the additional information these would provide.
L 263: In P. machaon it was shown that the diet changed its sensory response profile and also the neural code used for discrimination, making it more similar to that of P hospiton and less to that of P. machaon raised on a different diet. But this is not discussed by the authors
Reply: The results the reviewer describes refer to details about one lepidopteran species whereas many more references are available that would be just as well worthwhile to cite but would go beyond the scope of the Discussion.
L 279-280: I don't think the larvae discriminate in this case. Authors should rewrite the sentence
Reply: We have rephrased the sentence.
L 290-291: Similar results were found in P. hosption, a practically monophagous species, compared to the polyphagous P. machaon, with which they are closely related.
Reply: We do not concur that this additional citation is warranted because (1) the study on the two Papilio species is not about habituation; (2) Papilio machaon is an oligophagous species according to the generally accepted definition of oligophagy of herbivorous insects (e.g. Schoonhoven, van Loon and Dicke (2005) Insect-Plant Biology. Oxford University Press, 2nd Ed. 421 pp.).
L 331-334: this concept is very interesting, but it should be better argued, perhaps with some references
Reply: We rephrased the sentence to the extent that the two mechanisms mentioned that could contribute to deterrent neuron desensitisation are currently hypothetical.
L 345: I do not agree with this: in fact in the maxillary palp the greatest number of sensilla are mainly olfactory
Reply: The paper cited (Glendinning et al., 1998) states that the maxillary palps of Manduca sexta contain over 65% of the taste receptor cells of the caterpillars of this species and were demonstrated to have had a decisive role in mediating deterrency-based rejection of host plant disks treated with a non-host-plant extract. To our knowledge, the role of the maxillary palps in Helicoverpa (or Papilio) have not been studied and therefore cannot be excluded.
Reviewer 3 Report
This paper describes a strong parallel between physiological changes in taste neuron sensitivity and feeding habituation to the alkaloid strychnine in two species of closely related insect herbivores species, namely Helicoverpa armigera, a generalist feeder, and Helicoverpa assulta, a specialist feeder. The study was carried out using a behavioral bioassay (i.e., dual-choice leaf disk) bioassay and an extracellular tip recording technique. The results clearly show that caterpillars subjected to different durations of dietary exposure show that H. armigera did not habituate to the alkaloid deterrent after exposure after 24 h, 36 h, and 48 h, whereas, H. assulta displayed habituation after 48 h. It was not until 72 h that that H. armigera displayed habituation. These results agreed with electrophysiological results in that the deterrent-sensitive neuron in the galeal medial styloconic sensillum of both species displayed a reduced sensitivity to the deterrent.
General Comments:
This is a well-written paper. The authors have analyzed the results thoroughly and made logical conclusions based on their results and in keeping with previous findings in the literature. The figures are of good quality and thorough to explain their results. I have some specific comments, below.
Specific Comments:
- 55- insert “are” before “equipped” and change “equip” to “equipped”
- 62- insert “serve as” after “stimuli”
L 124-125- For the behavioral bioassays, the authors mentioned that they applied an aqueous concentration of strychnine to pepper plants to create disks and mentioned that such disks absorb such as solution evenly due to the “dark tissue” (line 129). Most leaf surfaces of such plants bear a hydrophobic upper layer and would homogeneous a distribution of the alkaloid during application? Why was not a weak alcoholic solution not used (i.e., as a diluent)?
L131-Rewrite—“disks were served”? Is “served” the correct word, here? You cannot change the thickness of the leaf!
L132- remove “makes” and replace with “so that”; change “can” to “could”
L132- if the strychnine solution was able to penetrate into the leaf (i.e., pepper disk), how did you know that it did not penetrate the entire thickness and leach out to the moist filter paper?
Author Response
This paper describes a strong parallel between physiological changes in taste neuron sensitivity and feeding habituation to the alkaloid strychnine in two species of closely related insect herbivores species, namely Helicoverpa armigera, a generalist feeder, and Helicoverpa assulta, a specialist feeder. The study was carried out using a behavioral bioassay (i.e., dual-choice leaf disk) bioassay and an extracellular tip recording technique. The results clearly show that caterpillars subjected to different durations of dietary exposure show that H. armigera did not habituate to the alkaloid deterrent after exposure after 24 h, 36 h, and 48 h, whereas, H. assulta displayed habituation after 48 h. It was not until 72 h that that H. armigera displayed habituation. These results agreed with electrophysiological results in that the deterrent-sensitive neuron in the galeal medial styloconic sensillum of both species displayed a reduced sensitivity to the deterrent.
General Comments:
This is a well-written paper. The authors have analyzed the results thoroughly and made logical conclusions based on their results and in keeping with previous findings in the literature. The figures are of good quality and thorough to explain their results. I have some specific comments, below.
Specific Comments:
- 55- insert “are” before “equipped” and change “equip” to “equipped”
Reply: Done
- 62- insert “serve as” after “stimuli”
Reply: We did not follow the advice given, rather clarified the sentence by inserting ‘defined as’ after ‘stimuli’.
L 124-125- For the behavioral bioassays, the authors mentioned that they applied an aqueous concentration of strychnine to pepper plants to create disks and mentioned that such disks absorb such as solution evenly due to the “dark tissue” (line 129). Most leaf surfaces of such plants bear a hydrophobic upper layer and would homogeneous a distribution of the alkaloid during application? Why was not a weak alcoholic solution not used (i.e., as a diluent)?
Reply: The pepper disks were cut from pepper fruit, therefore they did not have a hydrophobic upper layer and the solution did not run off or leach out. This explanation had already been added in response to comments of Reviewer 2. We have revised these lines to become more concise, see L130-135 of the revised manuscript.
L131- Rewrite—“disks were served”? Is “served” the correct word, here? You cannot change the thickness of the leaf!
Reply: Rephrased; see L130-135.
L132- remove “makes” and replace with “so that”; change “can” to “could”
Reply: The rephrasing and deletions described above have led to removal of the words indicated in this comment.
L132- if the strychnine solution was able to penetrate into the leaf (i.e., pepper disk), how did you know that it did not penetrate the entire thickness and leach out to the moist filter paper?
Reply: See our reply to the comments above.
Round 2
Reviewer 2 Report
No comments